∂ | Open Peer Review | Human Microbiome | Research Article

# Gut microbiota-derived extracellular vesicles form a distinct entity from gut microbiota

Anna Kaisanlahti,[1,2] Jenni Turunen,[1,3] Jenni Hekkala,[1,2] Surbhi Mishra,[1,2] Sonja Karikka,[1,2,4] Sajeen Bahadur Amatya,[1,2] Niko Paalanne,[3,5] Johanna Kruger,[6,7,8] Anne M. Portaankorva,[6,9] Jussi Koivunen,[10] Arja Jukkola,[11] Pia Vihinen,[12] Päivi Auvinen,[13,14] Sirpa Leppä,[15,16] Peeter Karihtala,[10,15] Vesa Koivukangas,[3] Janne Hukkanen,[17,18] Seppo Vainio,[4,19] Anatoliy Samoylenko,[4] Genevieve Bart,[4] Leo Lahti,[20] Justus Reunanen,[1,2] Mysore V. Tejesvi,[1] Terhi Ruuska-Loewald[1,3,5]

**ABSTRACT**   Extracellular vesicles (EVs), nanoparticles secreted by both gram-negative and gram-positive bacteria, carry various biomolecules and cross biological barriers. Gut microbiota-derived EVs are currently being investigated as a communication mechanism between the microbiota and the host. Few clinical studies, however, have investigated gut microbiota-derived EVs. Here, we show that machine learning models were able to accurately distinguish gut microbiota and respective microbiota-derived EV samples according to their taxonomic composition both within each data set (area under the curve [AUC] 0.764–1.00) and in a cross-study setting (AUC 0.701–0.997). These results show that gut microbiota-derived EVs form a distinct taxonomic entity from gut microbiota. Thus, conventional gut microbiota composition may not correctly reflect communication between the gut microbiota and the host unless microbiota-derived EVs are reported separately.

**IMPORTANCE**   Gut microbiota-derived extracellular vesicles (EVs) have been suggested to be a communication mechanism between the gut microbiota and the human body. However, the data on EV secretion from the gut microbiota remain limited. To investigate and compare the composition of gut microbiota-derived EVs to gut microbiota composition, we used a machine learning approach to classify 16S rRNA gene sequencing data in seven clinical data sets incorporating both gut microbiota and gut microbiota-derived EV samples. The results of the study show that microbiota-derived EVs form a separate taxonomic entity from the gut microbiota. Gut microbiota-derived EVs should be included in clinical studies that investigate gut microbiota to gain more comprehensive insight into gut microbiota–host communication.

**KEYWORDS**   extracellular vesicles, gut microbiota, gut microbiome, gut, bacteria, 16S RNA

Address correspondence to Anna Kaisanlahti , anna.kaisanlahti@oulu.fi.

Anna Kaisanlahti and Jenni Turunen contributed equally to this article. Author order was determined and agreed upon based on contribution.

Justus Reunanen, Mysore V. Tejesvi, and Terhi Ruuska-Loewald contributed equally to this article.

The authors declare no conflict of interest.

See the funding table on p. 15.

The human gut microbiome has been studied compositionally (1) and functionally (2) due to its contributions to human health. Changes in gut microbiota composition have been reported in gut-related pathologies including inflammatory bowel disease (IBD) (3) and colorectal cancer (4) but also in other pathologies such as type 1 diabetes (5) and neuropsychiatric disorders (6). Moreover, it has been suggested that gut microbiota plays a role in the onset of several diseases (7, 8). How changes in gut microbiota composition link to the pathologies outside the gut remains largely unresolved.

Extracellular vesicles (EVs) are lipid bilayer-coated nanoparticles secreted by both gram-negative and gram-positive bacteria. EVs carry various biomolecules as their cargo, including nucleic acids, proteins, lipids, and metabolites. Gut microbiota-derived EVs are

currently being studied as a communication mechanism between the gut microbiota and the body, as they can cross biological barriers and translocate via circulation to distal organs (9). However, few studies to date have explored gut microbiota-derived EVs as an integral part of gut microbiota analysis (10), despite accumulating evidence of their role in microbiota–host communication.

Our hypothesis was that microbiota-derived EVs may not directly represent the taxonomic composition of gut microbiota. We set out to investigate the differences in taxonomic compositions between gut microbiota-derived EVs and the respective gut microbiota samples in seven clinical data sets, six from our laboratory and one identified by systematic review. We used machine learning to classify gut microbiota-derived EV and gut microbiota samples first within each data set and then in a cross-study setting to test whether the classification was generalizable across the data sets.

## RESULTS

We first isolated EVs from fecal samples from six different clinical study cohorts, including patients with lymphoma ($n = 27$), Parkinson's disease ($n = 20$), solid tumors ($n = 28$) (11), obesity ($n = 30$) (12), and cohorts with pregnant women ($n = 23$) (13) and non-diseased volunteers ($n = 28$). We used 16S rRNA gene sequencing on fecal samples to identify the gut microbiota composition (referred to as FE in results) and on EVs isolated from fecal samples (referred to as EVs in results) to identify the originating taxa of the gut microbiota-derived EVs. Additionally, we performed a systematic literature review and identified one published data set with available taxonomic data of gut microbiota-derived EVs and gut microbiota composition from patients with colorectal cancer ($n = 70$) (14). Clinical and technical information on the data sets can be found in Table 1. Additional clinical data for the cohorts is presented in Table S1. Search queries used in the systematic review and Preferred Reporting Items for Systematic Reviews and Meta-Analyses flowchart of the systematic review process are presented in Table S2; Fig. S1. Additionally, lymphoma data set was further divided into subgroups including diffuse large B-cell lymphoma ($n = 15$) and Hodgkin's lymphoma ($n = 6$).

First, we pooled all gut microbiota and gut microbiota-derived EV samples from all the different cohorts and analyzed the differentially abundant genera between the gut microbiota and gut microbiota-derived EVs. In pooled samples, analysis of compositions of microbiomes with bias correction (ANCOM-BC) identified 78 taxa that were differentially abundant between sample groups. From these, 64 taxa were enriched and 14 depleted in gut microbiota-derived EVs as compared to gut microbiota (Fig. S2). In beta diversity analysis using principal coordinate analysis (PCoA), gut microbiota and gut microbiota-derived EV samples formed distinct clusters (Fig. 1). The beta diversity was significantly different between pooled gut microbiota-derived EVs and pooled gut microbiota samples when tested with permutational multivariate analysis of variance (PERMANOVA) ($P < 0.005$).

We then analyzed the taxonomy (Fig. 2; Tables S3 to S11) and differentially abundant genera between paired gut microbiota and gut microbiota-derived EVs in each data set. Depending on the data set, 4–157 differentially abundant taxa between the gut microbiota and gut microbiota-derived EV sample groups were identified (Table 2; Fig. S3 to S11).

We visualized the differences in beta diversity by PCoA in each data set. Gut microbiota and gut microbiota-derived EV samples formed distinct clusters in eight data sets, the only exception being Hodgkin's lymphoma. Similarly, beta diversity was significantly different ($P < 0.05$) between gut microbiota composition and EVs in the eight data sets (Fig. 3).

Next, we investigated whether we could distinguish between gut microbiota and gut microbiota-derived EV samples using gradient boosting, a machine learning classifier commonly used with gut microbiota composition data (15). For the initial analysis, we performed gradient boosting classification within each data set. The classifier was able to accurately classify gut microbiota and gut microbiota derived EV samples in all of

**TABLE 1** Clinical and technical data of 16S rRNA gene data sets[a]

| Data set | N | Laboratory | Sex female n (%) | Age mean (SD) | EV isolation method | EV characterization | Analyte in sequencing for EVs | Analyte in sequencing for feces | DNA isolation | Sequencing platform | BioProject ID |
|---|---|---|---|---|---|---|---|---|---|---|---|
| Lymphoma | 27 | Ruuska-Loewald[b] & Reunanen | 15 (54) | 61 (13) | Size exclusion and density gradient | TEM[c] NTA[d] | RNA | DNA | QIAamp Fast DNA Stool Mini Kit | IonTorrent | PRJNA1041060 |
| Parkinson's disease | 20 | Ruuska-Loewald & Reunanen | 9 (45) | 69 (3) | Size exclusion and density gradient | TEM NTA | RNA | DNA | QIAamp Fast DNA Stool Mini kit | IonTorrent | PRJNA1041975 |
| Solid tumor | 28 | Ruuska-Loewald & Reunanen | 9 (33) | 64 (10) | Size exclusion and density gradient | TEM NTA | RNA | DNA | QIAamp Fast DNA Stool Mini Kit | IonTorrent | PRJNA1020741 PRJNA1020742 |
| Pregnancy | 23 | Ruuska-Loewald & Reunanen | 23 (100) | 31 (5) | Size exclusion and density gradient | TEM NTA | RNA | DNA | DNeasy PowerSoil Pro Kit | IonTorrent | PRJNA878641 |
| Obesity | 30 | Ruuska-Loewald & Reunanen | 21 (70) | 47 (9) | Size exclusion and density gradient | TEM NTA | RNA | DNA | QIAamp Fast DNA Stool Mini Kit | IonTorrent | PRJNA1041054 |
| Non-diseased | 28 | Ruuska-Loewald & Reunanen | 17 (61) | 59 (14) | Size exclusion and density gradient | TEM NTA | RNA | RNA | QIAamp Fast DNA Stool Mini Kit | IonTorrent | PRJNA1020741 PRJNA1020742 PRJNA1041054 |
| Colorectal cancer | 70 | Park[e] | 24 (34) | 63 (9) | Differential centrifugation | None | DNA | DNA | DNeasy PowerSoil Pro kit | Illumina | PRJNA747730 |

[a]Clinical data includes the sex ratio and mean age, and technical data includes details on the laboratory workflow.
[b]Formerly Tapianen.
[c]Transmission electron microscopy.
[d]Nanoparticle tracking analysis.
[e]Used as the training data set in the cross-study.

the data sets with area under the curve (AUC) value describing classification accuracy ranging from 0.764 to 1.00 (Fig. 4).

The taxonomic features (genera) important for classification of gut microbiota and gut microbiota-derived EVs varied between the data sets. In lymphoma, *Staphylococcus* was the most important feature for classification. In Parkinson's disease, *Staphylococcus* and *Lactobacillus* were the two most important features. In patients with solid tumors, the most important features included *Rhodococcus* and *Allorhizobium-Neorhizobium-Pararhizobium-Rhizobium*. In pregnancy, multiple genera were important for the classification comprising *Porphyromonas*, *Lawsonella*, *Staphylococcus*, *Gemella*, *Alloprevotella*, *Peptoniphilus*, *Neisseria,* and *Finegoldia*. Similarly, in the obesity data set, multiple genera were important for the classification, including family *Oscillospiraceae* UGC-003, *Erysipelatoclostridiaceae*, *Listeria*, several identified and unidentified genera from the family *Lachnospiraceae*, *Clostridium sensu stricto 1*, *Oscillibacter*, *Streptococcus*, and *Parasutterella*. In the non-diseased cohort, the most important features included *Anaerostipes*, *Lactobacillus*, *Faecalibacterium*, family *Neisseriaceae*, *Fusobacterium,* and *Parasutterella*. The ten most important features for classification in each data set are presented in Fig. 5.

After each data set's classification was deemed accurate, we used the largest data set, colorectal cancer data set, to train a gradient boosting classifier and used this to classify the gut microbiota and gut microbiota-derived EV samples in the six other data sets. In this cross-study setting, the classifier accurately classified the gut microbiota-derived EVs and fecal samples in all data sets with AUC values ranging from 0.701 to 0.997 (Fig. 6). In addition, we verified the classification performance with three other classification algorithms for the within-data set classification and in a cross-study (Fig. S12 to S20), with mostly similar results.

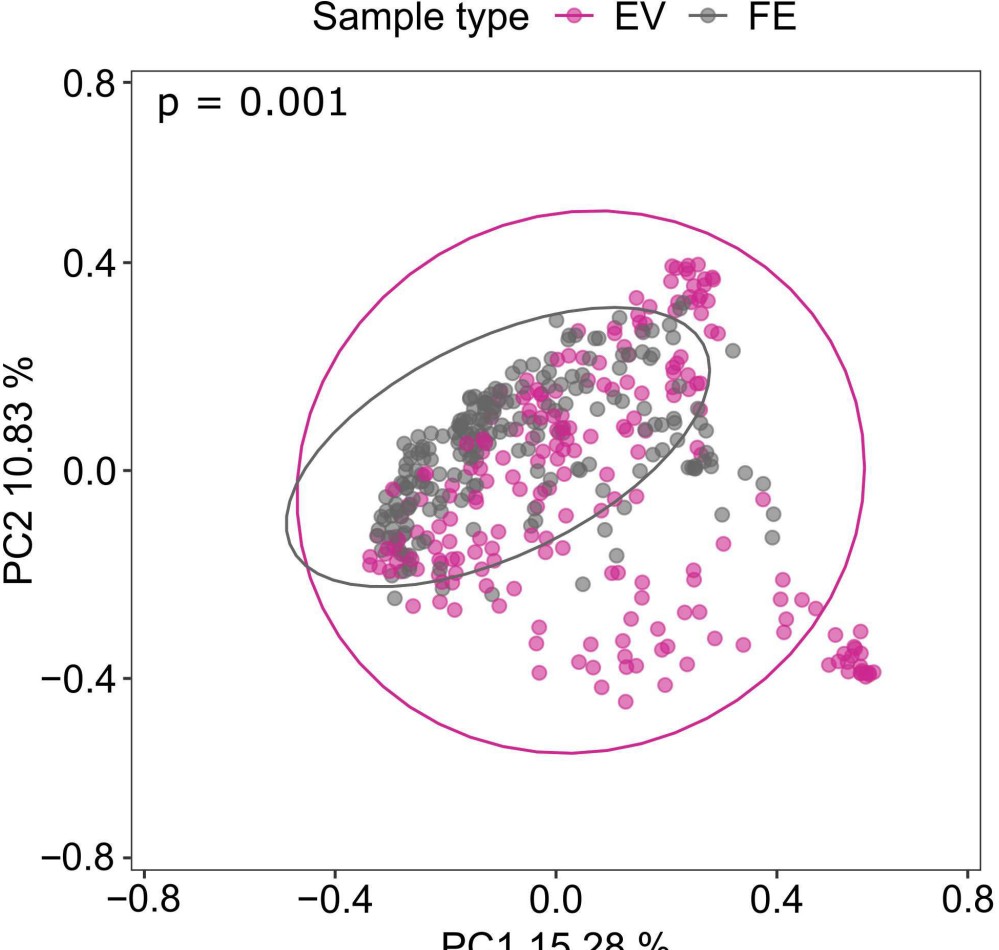

**FIG 1** Principal coordinate analysis with Bray–Curtis dissimilarity of gut microbiota and gut microbiota-derived extracellular vesicles. Gut microbiota-derived EV samples (EV) pooled from the different sample cohorts are presented as pink dots ($n = 221$). Gut microbiota samples (FE) pooled from each cohort are presented as grey dots ($n = 223$). Statistical significance was estimated with PERMANOVA. $P < 0.05$ was considered statistically significant.

## DISCUSSION

Using machine learning models with a cross-study design, we show that gut microbiota-derived EVs form a taxonomic entity distinct from gut microbiota composition in seven clinical data sets, six generated by us and one found by the systematic review. Despite the differences between EV isolation and sequencing methods, classification of gut microbiota-derived EVs and gut microbiota samples was accurate in the cross-study, strongly suggesting that the distinction between gut microbiota and gut microbiota-derived EV compositions is a generalized phenomenon. Thus, conventional gut microbiota composition may not correctly reflect communication between the gut microbiota and the host unless microbiota-derived EVs are reported separately. Our results suggest that gut bacteria may secrete EVs at different rates and that secretion activity is likely different between different taxa in different cohorts.

Several studies have investigated gut microbiota-derived EVs while exploring shifts in gut microbiota composition related to changes caused by clinical conditions (14, 16–22), probiotic use (23, 24), and dietary factors (25). However, these studies have usually not reported taxonomic differences between gut microbiota and EV compositions. Notably, Rodriguez-Diaz et al. (20) recently described modulation in EV secretion from gut microbiota in subjects with morbid obesity, diarrhea, and Crohn's disease,

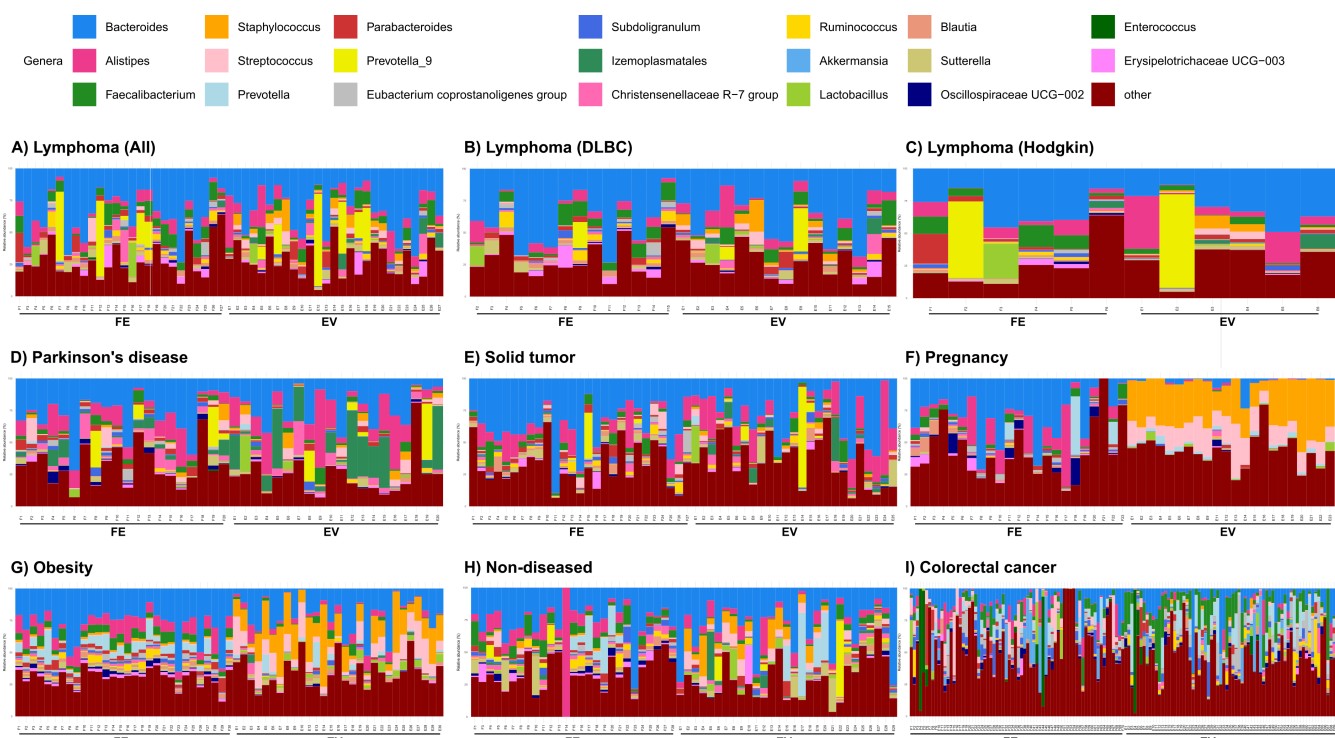

**FIG 2** The 20 most abundant genera in the gut microbiota-derived EVs and fecal samples in each data set, with the remainder assigned to the "other" category. (A) Lymphoma ($n = 53$, $n_{EV} = 27$, $n_{feces} = 26$); (B) lymphoma, diffuse large B-cell lymphoma ($n = 29$, $n_{EV} = 15$, $n_{feces} = 14$); (C) Hodgkin's lymphoma ($n = 12$, $n_{EV} = 6$, $n_{feces} = 6$); (D) Parkinson's disease ($n = 40$, $n_{EV} = 20$, $n_{feces} = 20$); (E) solid tumor ($n = 52$, $n_{EV} = 25$, $n_{feces} = 27$); (F) pregnancy ($n = 45$, $n_{EV} = 22$, $n_{feces} = 23$); (G) obesity ($n = 59$, $n_{EV} = 29$, $n_{feces} = 30$); (H) non-disease ($n = 55$, $n_{EV} = 28$, $n_{feces} = 27$); (I) colorectal cancer ($n = 140$, $n_{EV} = 70$, $n_{feces} = 70$).

concluding that gut microbiota-derived EVs modulate intestinal permeability in a way that depends on the subject's pathology. Furthermore, there are limited data on the composition of microbiota-derived EVs on the healthy human gut microbiota. Recently, Li et al. (10) characterized gut microbiota-derived EVs from five healthy individuals (10). Similarly to our findings in the non-diseased cohort, gut microbiota-derived EV composition varies from that of gut microbiota with lot of variation between individual samples. Comparing the taxonomic origin of EVs in healthy donors, Li et al. (10) report EVs to originate partially from the same genera indicated by the present

**TABLE 2** Table of the number of enriched and depleted genera in gut microbiota-derived EVs and gut microbiota samples all data sets according to ANCOM-BC

| Data set | Genera enriched in EVs (depleted in feces) | Genera depleted in EVs (enriched in feces) | Genera differentially abundant (total) |
|---|---|---|---|
| Pooled data | 64 | 14 | 78 |
| Lymphoma (all samples) | 8 | 14 | 22 |
| Lymphoma (diffuse large B cell lymphoma [DLBCL]) | 8 | 6 | 14 |
| Lymphoma (Hodgkin) | 4 | 3 | 7 |
| Parkinson's disease | 3 | 1 | 4 |
| Solid tumor | 7 | 34 | 41 |
| Pregnancy | 67 | 33 | 100 |
| Obesity | 84 | 73 | 157 |
| Non-diseased | 28 | 9 | 37 |
| Colorectal cancer | 11 | 24 | 35 |

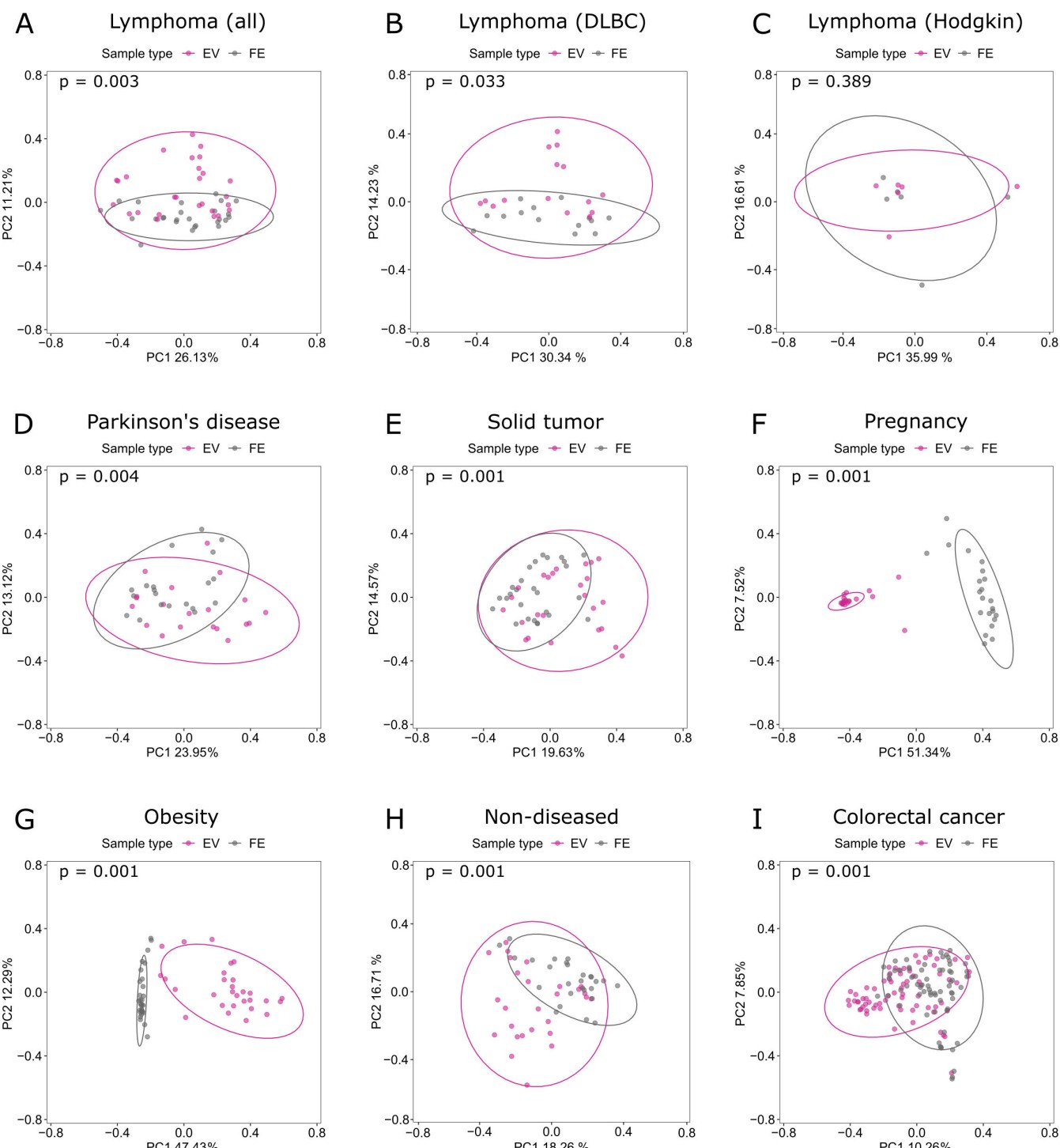

**FIG 3** Principal coordinate analysis with Bray–Curtis dissimilarity of gut microbiota and gut microbiota-derived extracellular vesicles in seven different clinical cohorts. (A) Lymphoma; (B) diffuse large B-cell lymphoma (subset); (C) Hodgkin's lymphoma (subset); (D) Parkinson's disease; (E) solid tumor; (F) pregnancy; (G) obesity; (H) non-diseased; (I) colorectal cancer. Each gray dot indicates one fecal (FE) microbiota sample. Each purple dot indicates the respective gut microbiota-derived extracellular vesicle (EV) sample isolated from the fecal samples. Statistical significance was estimated with PERMANOVA. *P* < 0.05 was considered statistically significant.

study, including *Bacteroides*, *Lactobacillus*, *Prevotella*, *Faecalibacterium,* and *Roseburia*. However, there are also differences in taxa of EV origin, most likely due to differences in sequencing methods and number of participants. Rodrigues-Diaz et al. (20) also

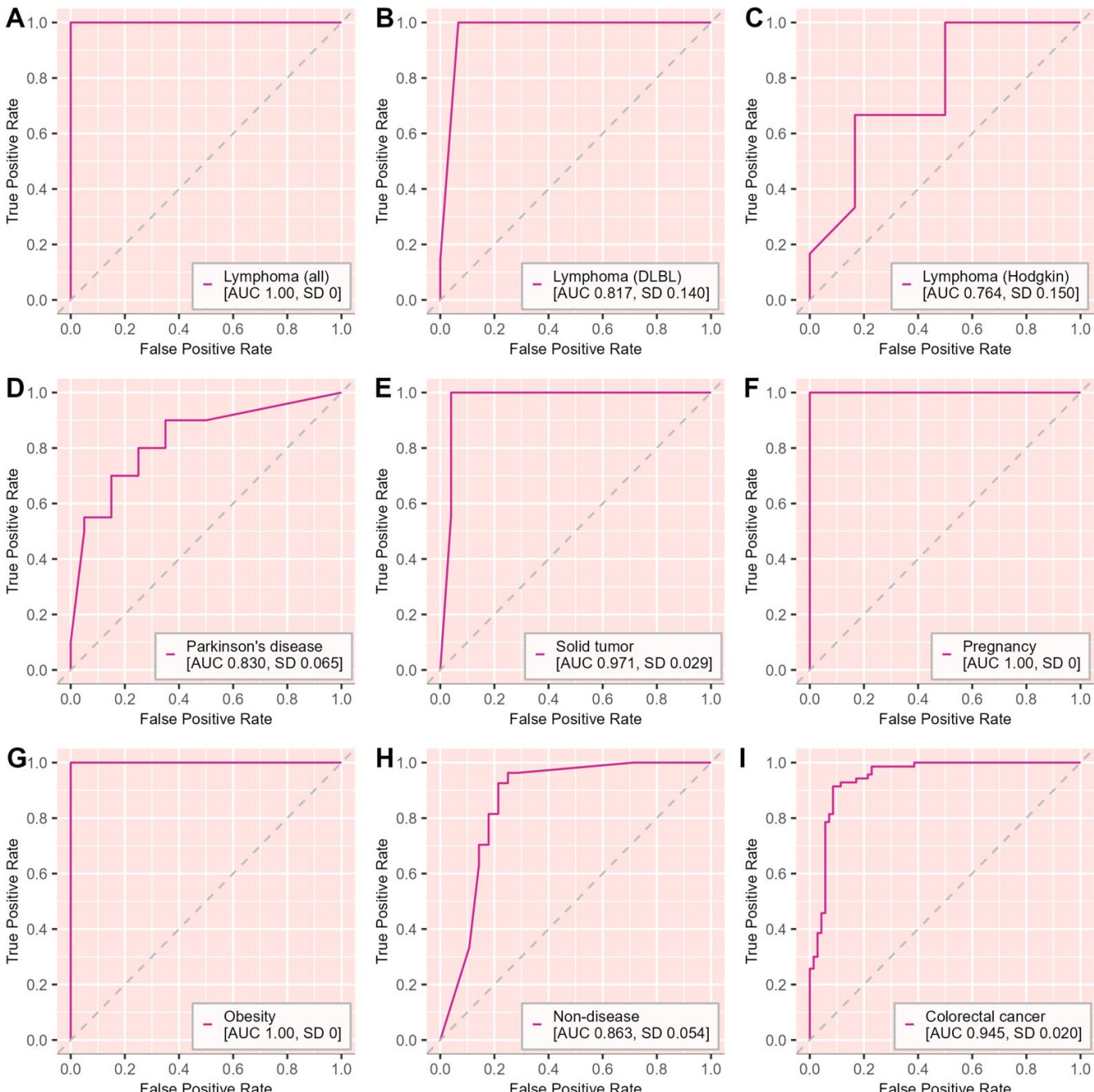

**FIG 4** Machine learning classification of gut microbiota and paired gut microbiota-derived EV samples within each data set using gradient boosting classifier. (A) Lymphoma ($n = 53$, $n_{EV} = 27$, $n_{feces} = 26$); (B) lymphoma, diffuse large B-cell lymphoma ($n = 29$, $n_{EV} = 15$, $n_{feces} = 14$); (C) Hodgkin's lymphoma ($n = 12$, $n_{EV} = 6$, $n_{feces} = 6$); (D) Parkinson's disease ($n = 40$, $n_{EV} = 20$, $n_{feces} = 20$); (E) solid tumor ($n = 52$, $n_{EV} = 25$, $n_{feces} = 27$); (F) pregnancy ($n = 45$, $n_{EV} = 22$, $n_{feces} = 23$); (G) obesity ($n = 59$, $n_{EV} = 29$, $n_{feces} = 30$); (H) non-disease ($n = 55$, $n_{EV} = 28$, $n_{feces} = 27$); (I) colorectal cancer ($n = 140$, $n_{EV} = 70$, $n_{feces} = 70$). The classification accuracy is presented as area under the curve (AUC) of receiver operating characteristic (ROC) figures. Standard deviation (SD) for AUC values is indicated in each data set. Dashed lines indicate random classification performance.

incorporated healthy controls ($n = 9$) into their study and identified five different genera differentially abundant between gut microbiota and gut microbiota-derived EVs. These included *Phascolarctobacterium*, *Veillonella*, and *Veillonellaceae_ge* as taxa depleted in gut microbiota-derived EVs and *Rikenellaceae_RC9_gut_group* and *Pseudomonas* as

## A) Lymphoma (all)

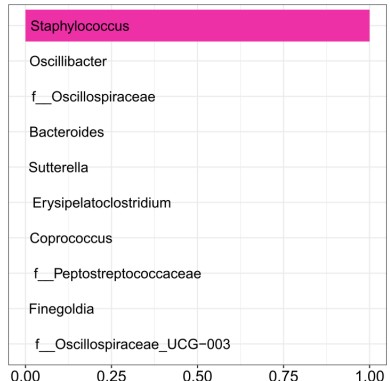

## B) Lymphoma (DLBC)

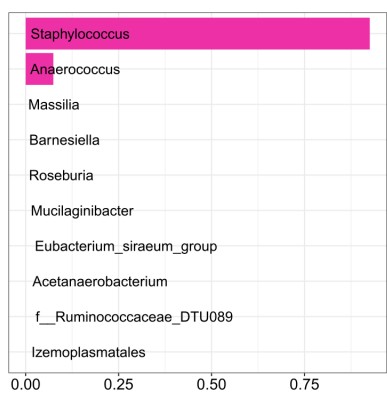

## C) Lymphoma (Hodgkin)

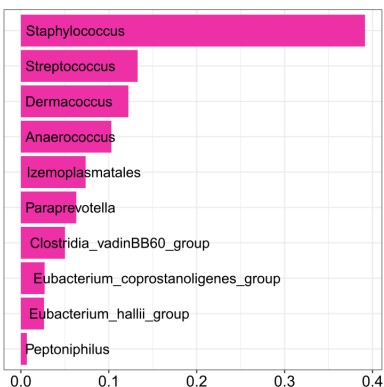

## D) Parkinson's disease

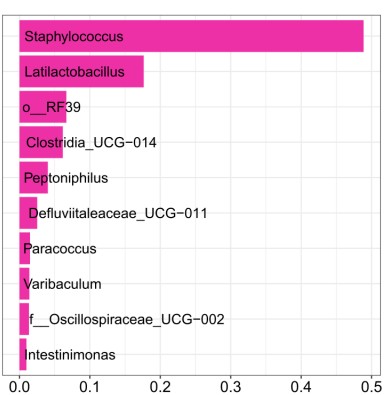

## E) Solid tumor

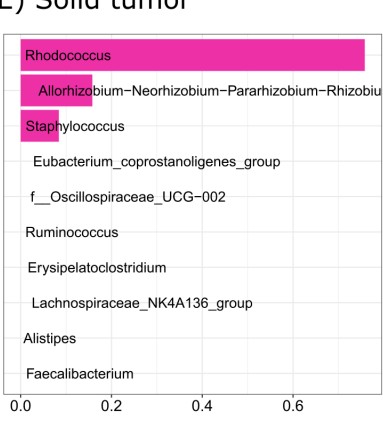

## F) Pregnancy

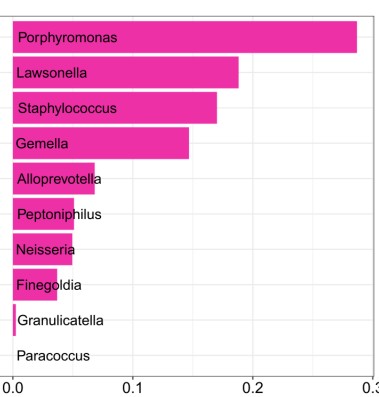

## G) Obesity

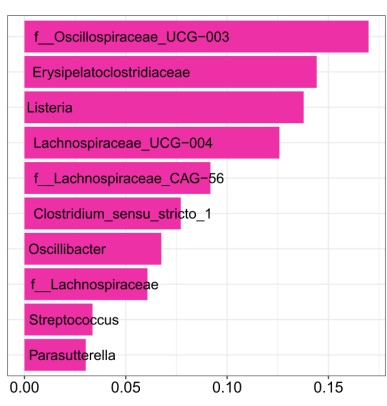

## H) Non-diseased

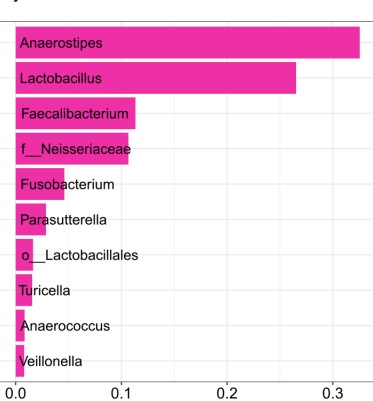

## I) Colorectal cancer

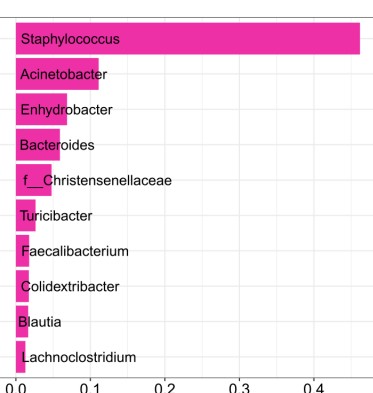

**FIG 5** The 10 most important taxonomic features for gradient boosting classification of gut microbiota and gut microbiota-derived EV samples in each data set. Feature importances are presented as importance scores for each data set. *Allorhizobium-Neorhizobium-Pararhizobium-Rhizobium.

enriched in gut microbiota-derived EVs. None of these taxa were abundant in the gut microbiota-derived EVs of non-diseased participants in our study.

Different cancers (26–30), Parkinson's disease (31), obesity (32, 33), and pregnancy (34) are all associated with changes in gut microbiota composition. In general, most of the taxa that were differentially abundant between gut microbiota and gut microbiota-derived EVs were enriched in the EVs as shown by analysis with pooled samples. However, there were a lot of differences between individual clinical cohorts. In the three cancer data sets, lymphoma, solid tumors, and colorectal cancer, there were numerous

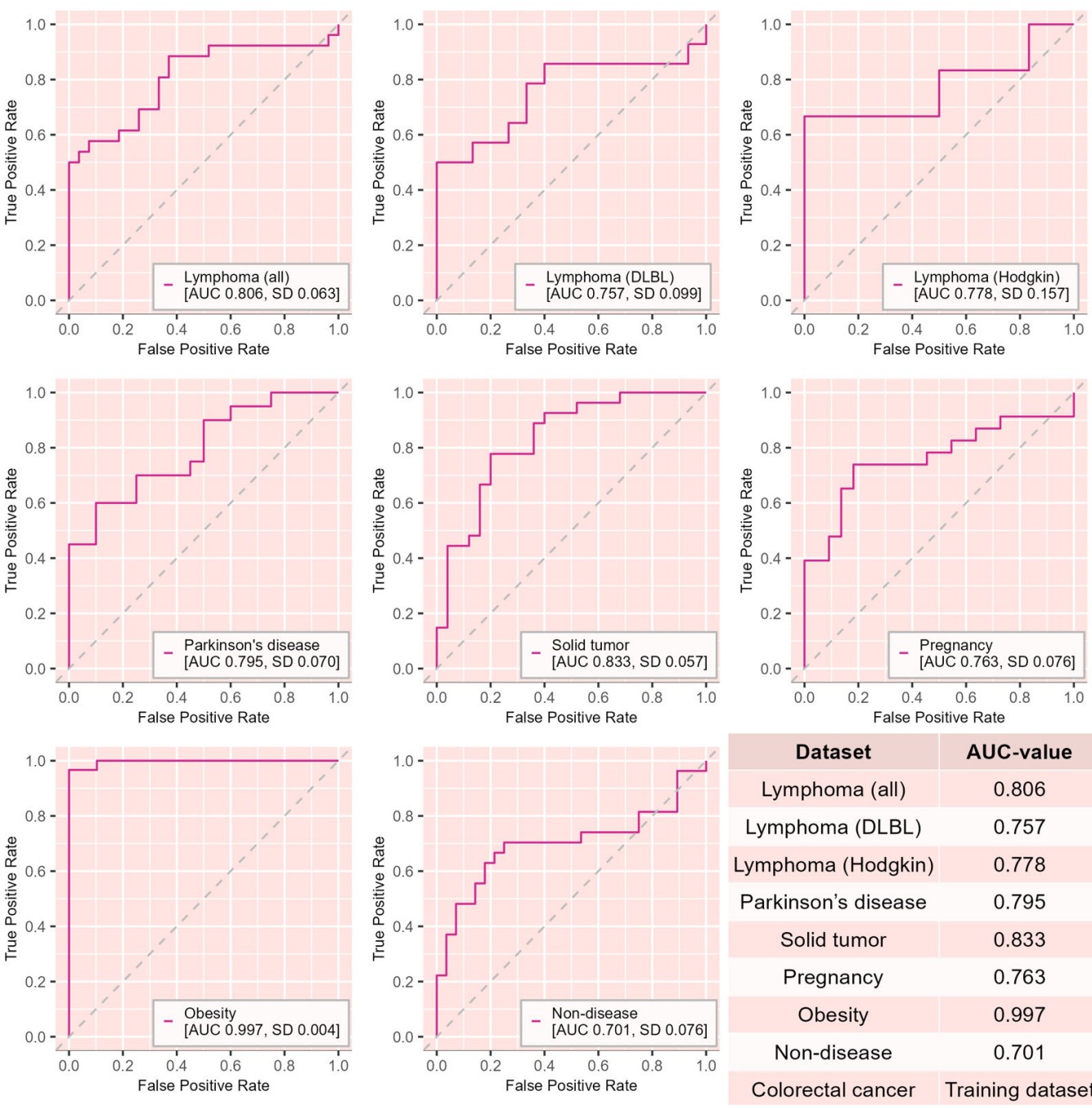

**FIG 6** Cross-study machine learning classification of gut microbiota and paired gut microbiota-derived EV samples using gradient boosting classifier. (A) Lymphoma ($n = 53$, $n_{EV} = 27$, $n_{feces} = 26$); (B) lLymphoma, diffuse large B-cell lymphoma ($n = 29$, $n_{EV} = 15$, $n_{feces} = 14$); (C) Hodgkin's lymphoma ($n = 12$, $n_{EV} = 6$, $n_{feces} = 6$); (D) Parkinson's disease ($n = 40$, $n_{EV} = 20$, $n_{feces} = 20$); (E) solid tumor ($n = 52$, $n_{EV} = 25$, $n_{feces} = 27$); (F) pregnancy ($n = 45$, $n_{EV} = 22$, $n_{feces} = 23$); (G) obesity ($n = 59$, $n_{EV} = 29$, $n_{feces} = 30$); (H) non-disease ($n = 55$, $n_{EV} = 28$, $n_{feces} = 27$); (I) colorectal cancer ($n = 140$, nEV = 70, $n_{feces} = 70$). The classification accuracy is presented as area under the curve (AUC) of receiver operating characteristic (ROC) figures. Standard deviation (SD) for AUC values is indicated in each data set. Dashed lines indicate random classification performance.

taxonomic groups that were depleted in EVs as compared to gut microbiota. In contrast to this, in the non-diseased group, more taxa were enriched in EVs as compared to gut microbiota samples. When comparing differentially abundant genera to the taxa important for the sample classification, there was a lot of variation if the important features were enriched or depleted in EVs. For example, in the obesity data set, most

of the important features for classification were depleted in EVs as compared to feces. In contrast, most of the features important for classification were enriched in EVs in pregnancy, colorectal cancer, and non-diseased data sets. Taken together, the differences between clinical cohorts suggest that the EV secretion from gut microbiota is modulated in different clinical conditions beyond the changes observed in gut microbiota composition. This modulation might be more pronounced in some conditions, such as pregnancy and obesity, in which the taxonomic differences between gut microbiota and gut microbiota-derived EVs were more distinct. Studying disease-specific modulations on gut microbiota-derived EVs could provide important insight into how gut microbiota plays a part in the pathology in different diseases.

In this study, using data sets produced by us, we sequenced DNA from fecal samples to assess the gut microbiota composition and RNA from EVs to trace their origin. This approach was taken because the presence of genomic DNA in bacterial EVs remains understudied with no consensus (35–37), whereas the presence of RNA in bacterial EVs is well documented. For example, bacterial small RNA packaged in EVs has been observed to have a functional role in bacteria-host interactions, such as immune system regulation by particularly pathogenic bacteria (38), although the functions of specifically 16S rRNA packaged into EVs remain unknown. In the colorectal cancer data set included in the analysis (14), DNA was sequenced from gut microbiota-derived EVs. Despite differences in the use of RNA and DNA, we obtained similar results in classification accuracy in all of the data sets regardless of which molecule was sequenced.

Our results suggest that gut microbiota-derived EVs represent a taxonomically separate layer of gut microbiota that may mediate host–microbiota communication. Our results challenge the idea that EVs secreted by gut microbiota correspond directly to the gut microbiota composition (39–42). Interestingly, previous studies have suggested that circulating gut microbiota-derived EVs could be better markers in clinical conditions such as IBD or colorectal cancer than the gut microbiota composition (14, 17). Investigating gut microbiota-derived EVs with metagenomics or other approaches could reveal much novel cross talk via EVs between the gut microbiota and the host. For these reasons, gut microbiota-derived EVs should be actively investigated as distinct entities in clinical microbiome research.

This study has several strengths. Machine learning classifiers verified the differences between paired gut microbiota and gut microbiota-derived EV samples in seven clinical data sets. Although there were marked variations across different cohorts, including different EV isolation techniques and analysis methods (Table 1), accurate classification of gut microbiota-derived EVs and fecal samples was a universal finding in all data sets. This was further confirmed by a cross-study design which confirmed that the main finding was generalizable across the data sets. Furthermore, we used RNA-based identification for microbiota-derived EVs in our data sets due to plenty of evidence of the presence of RNA in these vesicles. In the cross-study setting, we built a gradient boosting machine learning model using the colorectal cancer data set by Park et al. (14), where DNA-based bacterial identification was used for both gut microbiota and gut microbiota-derived EV samples (14). This model successfully classified samples in data sets where RNA-based identification was used for EV samples. Although using DNA in fecal samples and RNA in EVs could introduce potential bias in the analysis, our finding suggests that the compositional differences between gut microbiota and gut microbiota-derived EVs persist and are not caused by the use of DNA or RNA in EV sequencing.

The study also has some limitations. We were unable to retrieve all the published data sets identified in the systematic review and subsequently verify the classification performance because they were not deposited in publicly available repositories. Thus, one of the limitations of this work is the absence of a validation data set that is due to limited data availability on EVs in public repositories. Another limitation is the possible overfiltration of EVs caused by our chosen EV isolation method, for example, given the heterogeneity of bacterial EVs, use of ultracentrifugation might exclude a subpopulation of EVs. Moreover, the 16S rRNA gene sequencing approach might not be comprehensive

when analyzing the EV taxonomy of origin, while EVs without 16S rRNA are not detected. In addition, the efficiency of RNA to cDNA conversion might affect the obtained results. However, 16S rRNA gene sequencing remains the best option for studying the compositional origin of bacterial EVs. There is currently no standardized approach for 16S rRNA gene sequencing to analyze microbiota-derived EVs. While it has been suggested that full-length 16S rRNA sequencing would enable more comprehensive characterization of gut microbiota-derived EVs, most studies have sequenced V3–V4 or V4–V5 regions of the gene in this context. Further work is required to assess the characterization of EV origin using DNA and RNA in gut microbiota-derived EVs. Another limitation of this work is the risk of contamination: despite the use of negative controls and good laboratory practices, the chance of sample contamination affecting the obtained results cannot be excluded comprehensively.

In conclusion, these results suggest that gut microbiota-derived EVs form a taxonomic entity distinct from gut microbiota composition. Gut microbiota-derived EVs should likely be investigated and reported separately in clinical microbiome research to further evaluate their clinical importance. We suggest that a concept of "EV-biome" or "communicatome" (i.e., the characteristics of microbiota-derived EVs) be included in studies investigating gut microbiota–host communication.

## MATERIALS AND METHODS

### Collection of fecal samples from clinical cohorts

Fecal samples were obtained from patients/participants at Oulu University Hospital, Oulu, Finland, and Helsinki University Hospital, Helsinki, Finland, who provided their written informed consent. After collection, fecal samples were stored in −20°C or −80°C before further processing. Laboratory work was carried out at the University of Oulu. The clinical data sets comprised study subjects with lymphoma ($n = 27$), Parkinson's disease ($n = 20$), solid tumors ($n = 28$) (11), pregnancy ($n = 23$) (13), obesity ($n = 30$) (12), and data set with non-diseased participants ($n = 28$). Additional clinical information available for the clinical cohorts is provided in Table S1.

### EV isolation and characterization

We isolated EVs from fecal samples from six different clinical study populations. Bacteria-enriched EVs were isolated from fecal matter using size exclusion chromatography and density gradient ultracentrifugation as described previously (43). Briefly, feces were suspended in phosphate-buffered serum (PBS) and centrifuged to remove solid materials, followed by sample filtration. Filtrate was concentrated and EVs isolated using Exo-Spin Mini-Columns (Cell Guidance Systems). Obtained EVs were subjected to gradient density ultracentrifugation (OptiPrep), followed by collection and washing of density fractions 6 and 7. A negative control sample comprising PBS was included in the isolation protocol. Isolated EVs were characterized with nanoparticle tracking analysis (NTA) and transmission electron microscopy (TEM). NTA and TEM data are available for the previously published studies (11, 13). Detailed EV isolation and characterization methods are available as EV-track submission under EV-TRACK ID EV240156 (available at http://evtrack.org/review.php).

### Isolation of RNA and DNA, cDNA conversion, PCR, and 16S rRNA gene sequencing

Total RNA was isolated from the EVs using an exoRNeasy Serum Plasma Midi Kit (Qiagen, Germany) according to the manufacturer's protocol. RNA was reverse-transcribed to cDNA using iScript cDNA Synthesis Kit (BioRad, USA) according to the manufacturer's protocol. Total DNA was isolated from the fecal samples using a DNeasy PowerSoil Pro Kit (Qiagen) (pregnancy data set) or QIAamp Fast DNA Stool Mini Kit (Qiagen) according to the manufacturer's protocol.

The isolated DNA and cDNA went through PCR using 16S rRNA gene primers 519F (5′-CAGCMGCCCGCGGTAATWC-3′) and 926R (5′-CCGTCAATTCCTTTRAGTTT-3′) targeting the V4-V5 region of the gene (44, 45). Each of the primers included a unique barcode for identifying the samples after pooling them for sequencing. The PCR was performed using Phusion Flash High Fidelity PCR master mix (Thermo Scientific, USA) according to the manufacturer's protocol in duplicate 15-µL reactions (lymphoma, Parkinson's disease, solid tumor, and non-diseased data sets), duplicate 20-µL reactions (obesity data set), or a single 50-µL reaction (pregnancy data set) using Veriti 96-well thermal cycler (Thermo Scientific). For duplicate 15-µL reactions, the PCR program was as follows: 3 min at 98°C, 22 cycles of amplification at 64°C annealing temperature, and final elongation of 5 min at 72°C. For the 50-µL reaction, the PCR program was as follows: 3 min at 98°C, followed by 30 cycles of amplification at 56°C annealing temperature, and final elongation of 5 min at 72°C. Each reaction's success was confirmed by agarose gel electrophoresis.

The sequencing of the PCR products was performed with IonTorrent PGM (Thermo Fisher Scientific) by Biocenter Oulu Sequencing Core. The PCR products were pooled and purified by AMPure XP (Beckman Coulter, USA). The product went through agarose gel electrophoresis in 1% gel, and the resulting product was purified with MinELute Gel Extraction Kit (Qiagen). An additional PCR was performed for the pooled sample using 1-µM primers A and trP1, with a program that included seven cycles of annealing at 63°C. The pooled PCR product was purified again with AMPure XP, analyzed with Bioanalyzer, and concentration was measured with Quant-iT PicoGreen dsDNA Assay Kit (Thermo Fisher Scientific). After the concentration measurements, sequencing was performed using Ion PGM Hi-Q View Template Kit for 400-bp templating program, Ion PGM Hi-Q View Sequencing Kit for 850 cycles, and 316 v2 chip.

## Identification of additional data sets

To identify published 16S rRNA data sets, a systematic literature review was performed. PubMed, Scopus, and Web of Science were searched for publications examining gut microbiota and gut microbiota-derived EVs through 16S rRNA gene sequencing analysis. The search was planned to find publications incorporating one word from each of the following three categories in their titles or abstracts: (i) extracellular vesicle and outer membrane vesicle; (ii) bacteria, microbiota, and microbiome; and (iii) 16S, metagenome, next-generation sequencing, composition, and abundance (Table S2). The identified publications were imported into Covidence systematic review software (46). Three independent reviewers screened the abstracts and full text, excluding irrelevant publications and article types other than original research (Fig. S1). The inclusion criteria were samples of human origin, EVs of gut microbiota origin, and sequencing analysis incorporating both gut microbiota and gut microbiota-derived EVs.

## Sequence processing

The raw sequences of all data sets were prepared using QIIME2 (versions 2023.5 and 2023.9-amplicon) (47) according to a previous pipeline (13). In short, sequences shorter than 200 bp were removed, and the remaining sequences were demultiplexed. The demultiplexed sequences were denoised with parameters listed in Table S21. For data sets with available negative control samples, we removed likely contaminants using R package decontam (48). Finally, we removed samples with less than 100 reads from the analyses. For taxonomy analysis of the resulting amplicon sequence variants, we used the SILVA database (version 138) (49). We identified bacteria in each data set on the genus level and removed the following taxa as likely skin or environmental contaminants based on earlier research on how to deal with contamination (50), as well as non-bacteria: mitochondria, eukaryotes, *Cyanobacteria*, archaea, *Corynebacterium*, and *Cutibacterium*.

## Taxonomy and diversity analysis

We calculated relative bacterial abundances in the gut microbiota and gut microbiota-derived EV samples and performed differential abundance analysis using ANCOM-BC

## A    Sample processing

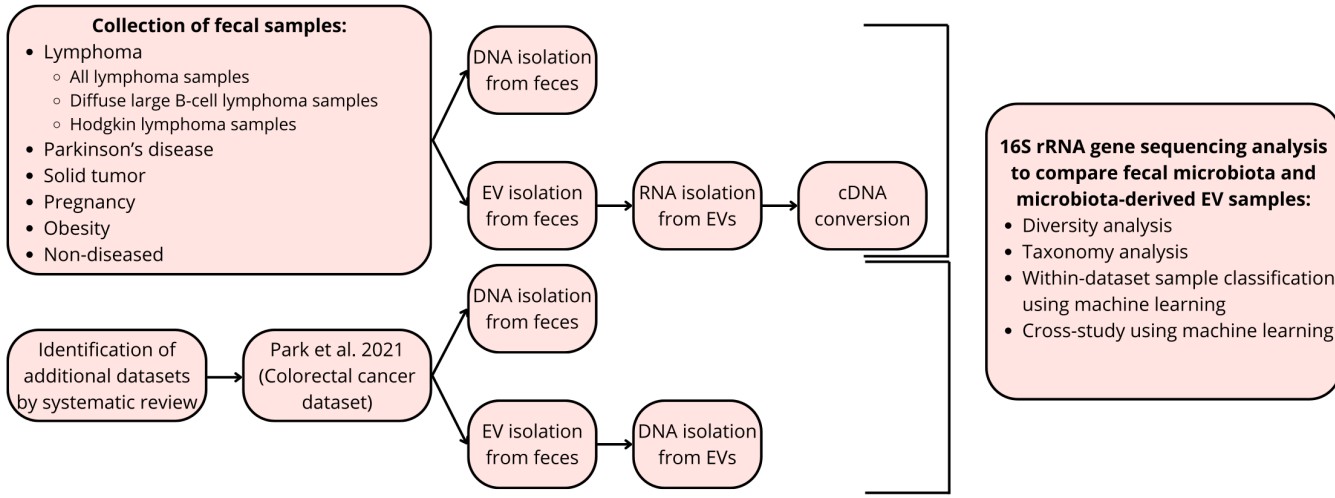

## B    Classification analysis using machine learning

### Within-dataset sample classification

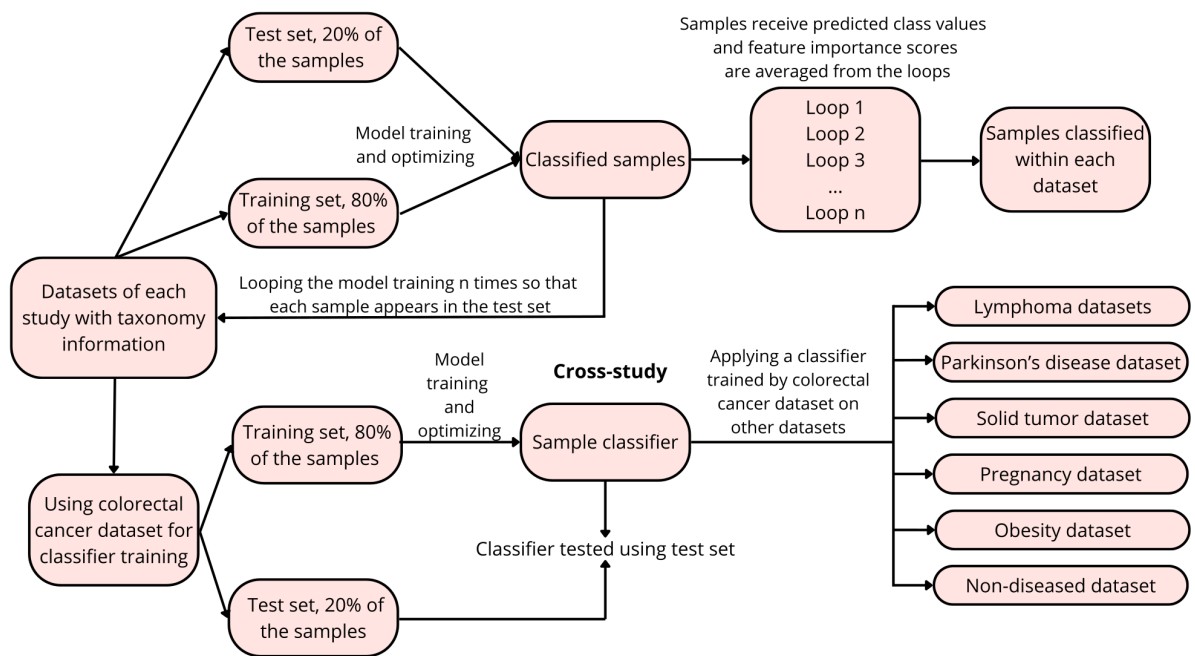

**FIG 7**   Flowchart of the study. (A) Study setting. (B) Machine learning analysis.

to explore differential genus-level abundance between the gut microbiota and gut microbiota-derived EVs (51). We visualized the beta diversity of the gut microbiota and gut microbiota-derived EVs with PCoA using Bray–Curtis dissimilarity and compared their statistical differences using PERMANOVA. For beta diversity analysis, all data sets were rarefied to even sampling depth based on the minimum sample read count, omitting samples with less than 100 reads. The read counts for each data set are presented in Tables S12 to S20.

## Machine learning analysis

First, we performed machine learning analysis within each data set and subsequently a cross-study machine learning analysis using a colorectal cancer data set for classifier training (Fig. 7). Both analyses were performed with QIIME2 with four different algorithms: gradient boosting and random forest were chosen while they are commonly used in gut microbiota research. Due to the novelty of gut microbiota-derived EVs as a research field, two other algorithms, extra trees and linear support vector classifier, were included in the analysis. The trained classifiers were used to classify whether the sample belonged to the gut microbiota or the gut microbiota-derived EV sample group. For the within-data set classification, we used nested cross-validation. In this approach, each data set was divided into training (80%) and test data sets (20%), followed by classifier training and testing. This process was looped until each sample appeared once in the test data set. The number of estimators was set to 100, and the number of k-fold cross-validations to perform was set to five. For the cross-study, the same four algorithms were trained with the largest data set (i.e., colorectal cancer) and used to classify samples from the other six clinical data sets. The colorectal cancer data set was divided into training (80%) and test data sets (20%). The number of estimators was set to 100, and the number of k-fold cross-validations to perform was set to five. The model was then used to classify the samples in the six other data sets. The ANCOM-BC figures were drawn in QIIME2. All other figures were drawn using R (version 4.3.1) packages reshape2 (version 1.4.4), plyr (version 1.8.8), dplyr (version 1.1.3), pROC (version 1.18.4), and ggplot2 (version 3.4.3).

## Statistical analysis

To explore the genus-level differential abundance between the gut microbiota and gut microbiota-derived EVs, we used ANCOM-BC. Differential abundance was considered significant at $P < 0.05$. The $P$-values were adjusted using the Bonferroni correction. In the beta diversity analysis, we used PERMANOVA with a significance threshold of $P < 0.05$.

## ACKNOWLEDGMENTS

We thank the Biocenter Oulu sequencing core facility for their expertise.

We thank the Academy of Finland (grants 328768, 299749, and 243032491 to J.R.), National Institutes of Health (grant R01AB123456 to T.R.L.), Pediatric Research Foundation (T.R.L.), State Funding for University Hospitals (VTR grant to T.R.L.), Finnish Cultural Foundation (grant 00220426 to A.K.), and Emil Aaltonen Foundation (grant 230067 to A.K.) for funding the project.

Conceptualization: J.R., M.V.T., and T.R.L. Methodology: A.K., J.T., J.H., S.V., A.S., G.B., L.L., and M.V.T. Data acquisition: N.P., J.K., A.P., J.K., A.J., P.V., P.A., S.L., P.K., V.K., J.H., and T.R.L. Experimental work: A.K., J.T., J.H., S.M., S.K., S.B.A., and T.R.L. Analysis of the data: A.K., J.T., J.H., L.L., M.V.T., and T.R.L. Interpretation of the results: A.K., J.T., J.H., L.L., J.R., M.V.T., and T.R.L. Writing of the manuscript: A.K., J.T., J.H., L.L., J.R., M.V.T., and T.R.L. Funding acquisition: J.R. and T.R.L.

## AUTHOR AFFILIATIONS

[1]Biocenter Oulu, University of Oulu, Oulu, Finland

[2]Research Unit of Translational Medicine, University of Oulu, Oulu, Finland

[3]Research Unit of Clinical Medicine, University of Oulu, Oulu, Finland

[4]Disease Networks Research Unit, University of Oulu, Oulu, Finland

[5]Department of Pediatrics and Adolescent Medicine, Oulu University Hospital, Oulu, Finland

[6]Research Unit of Clinical Medicine, Neurology, University of Oulu, Oulu, Finland

[7]Neurocenter, Neurology, Oulu University Hospital, Oulu, Finland

[8]Medical Research Center, Oulu University Hospital, Oulu, Finland

[9]Clinical Neurosciences, University of Helsinki, Helsinki, Finland

[10]Department of Medical Oncology and Radiotherapy and Medical Research Center, Oulu University Hospital, University of Oulu, Oulu, Finland

[11]Tampere Cancer Center, Tampere University, Tampere, Finland

[12]FICAN West Cancer Centre, Turku University Hospital, University of Turku, Turku, Finland

[13]Cancer Center, Kuopio University Hospital, The Wellbeing services county of North Savo, Kuopio, Finland

[14]Institute of Clinical Medicine, University of Eastern Finland, Kuopio, Finland

[15]Department of Oncology, Helsinki University Hospital Comprehensive Cancer Center, University of Helsinki, Helsinki, Finland

[16]Research Programs Unit, Applied Tumor Genomics, Faculty of Medicine, University of Helsinki, and iCAN Digital Precision Cancer Medicine Flagship, Helsinki, Finland

[17]Research Unit of Biomedicine and Internal Medicine, University of Oulu, Oulu, Finland

[18]Medical Research Center Oulu, University of Oulu, Oulu University Hospital, Oulu, Finland

[19]Kvantum Institute, University of Oulu, Oulu, Finland

[20]Department of Computing, University of Turku, Turku, Finland

## AUTHOR ORCIDs

Anna Kaisanlahti http://orcid.org/0000-0003-2778-5621

Jenni Turunen http://orcid.org/0000-0002-4158-7026

## FUNDING

| Funder | Grant(s) | Author(s) |
|---|---|---|
| Academy of Finland | 328768, 299749, 243032491 | Justus Reunanen |
| National Institutes of Health | R01AB123456 | Terhi Ruuska-Loewald |
| Pediatric Research Foundation | | Terhi Ruuska-Loewald |
| State Funding for University Hospitals | VTR grant | Terhi Ruuska-Loewald |
| Suomen Kulttuurirahasto | 00220426 | Anna Kaisanlahti |
| Emil Aaltosen Säätiö | 230067 | Anna Kaisanlahti |
| Päivikki and Sakari Sohlberg Foundation | | Mysore V. Tejesvi |

## DATA AVAILABILITY

The 16S rRNA gene sequencing data for the seven data sets are available from https://www.ncbi.nlm.nih.gov/ under the following BioProject identifiers: for fecal samples and EVs from patients with lymphoma, PRJNA1041060; for fecal samples and EVs from patients with Parkinson's disease, PRJNA1041975; for fecal samples and EVs from patients with solid tumors, PRJNA1020741 and PRJNA1020742; for fecal samples and EVs from participants with pregnancy, PRJNA878641; for fecal samples and EVs from patients with obesity, PRJNA1041054; for fecal samples and EVs from non-diseased participants, PRJNA1020741, PRJNA1020742, and PRJNA1041054; and for fecal samples and EVs from patients with colorectal cancer, PRJNA747730. The code used in this work is available at https://doi.org/10.5281/zenodo.14638120.

## ETHICS APPROVAL

The research protocols were approved by the Ethical Committee of the Northern Ostrobothnia Hospital District at Oulu University Hospital (EETTMK:96/2008, EETTMK:103/2018, EETTMK:3/2016, EETTMK:11/2019, EETTMK:12/2020), Finland, and the

Helsinki University Hospital District Regional Committee on Medical Research Ethics (HUS/1377/2020).

## ADDITIONAL FILES

The following material is available online.

### Supplemental Material

**Supplemental figures, part 1 (mSystems00311-25-s0001.pdf).** Figures S1 to S11.
**Supplemental figures, part 2 (mSystems00311-25-s0002.pdf).** Figures S12 to S20.
**Supplemental tables (mSystems00311-25-s0003.pdf).** Tables S1 to S21.

### Open Peer Review

**PEER REVIEW HISTORY (review-history.pdf).** An accounting of the reviewer comments and feedback.

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
