## [Reviewer comments · mSystems]

Gut microbiota-derived extracellular vesicles form a distinct entity from gut microbiota

Anna Kaisanlahti, Jenni Turunen, Jenni Hekkala, Surbhi Mishra, Sonja Karikka, Sajeen Amatya, Niko Paalanne, Johanna Krüger, Anne Portaankorva, Jussi Koivunen, Arja Jukkola, Pia Vihinen, Päivi Auvinen, Sirpa Leppä, Peeter Karihtala, Vesa Koivukangas, Janne Hukkanen, Seppo Vainio, Anatoliy Samoylenko, Genevieve Bart, Leo Lahti, Justus Reunanen, Mysore Tejesvi, and Terhi Ruuska-Loewald

Corresponding Author(s): Anna Kaisanlahti, Oulun yliopisto

Review Timeline:

Submission Date:

March 4, 2025

Accepted:

April 3, 2025

Editor: Gail Rosen

Reviewer(s): The reviewers have opted to remain anonymous.

Transaction Report:

DOI: <https://doi.org/10.1128/msystems.00311-25>

Re: mSystems00311-25 (Gut microbiota-derived extracellular vesicles form a distinct entity from gut microbiota)

Dear Mrs. Anna Kaisanlahti:

Your manuscript has been accepted, and I am forwarding it to the ASM production staff for publication. Your paper will first be checked to make sure all elements meet the technical requirements. ASM staff will contact you if anything needs to be revised before copyediting and production can begin. Otherwise, you will be notified when your proofs are ready to be viewed.

Sincerely,

Gail Rosen
Editor
mSystems

Reviewer #2 (Comments for the Author):

Okay, I do honestly thank the authors for putting up with my extensive criticisms and comments and doing their best to accommodate my comments. I think the MS is acceptable.

The expansion of technical limitations is reasonable and at least slightly addresses my issues. I appreciate that some of the issues are pervasive in the EV field -- particularly OMVs/bacterial EVs -- which is why I'm raising them. This isn't due to a technical problem per se; it is because many investigators are failing to even properly identify what the confounder are. The fact that so many confounders exist and then the RNA/DNA thing is just thrown on top without any control is just irksome to me. (As an aside, all of my comments about dead cells are not whether there are dead cells contaminating the preps, it is whether bacterial EVs don't represent a mixture of different small particles, some of which are the byproducts of cell death. This is extremely difficult to address in the bacterial world, though seems to be on better footing in the eukaryotic EV world now that EVs have been defined so specifically.

Regardless of any remaining criticisms, this MS represents a substantial piece of careful work, and I think the authors have mostly been circumspect in their conclusions. I thank them for their attention to my harsh reviews.

No further edits requested.